# Safety and efficacy study: Short-term application of radiofrequency ablation and stereotactic body radiotherapy for Barcelona Clinical Liver Cancer stage 0–B1 hepatocellular carcinoma

Feiqian Wang[1,2], Kazushi Numata [1]*, Atsuya Takeda[3], Katsuaki Ogushi[1], Hiroyuki Fukuda[1], Koji Hara[1], Makoto Chuma[1], Takahisa Eriguchi[3], Yuichirou Tsurugai[3], Shin Maeda[4]

1 Gastroenterological Center, Yokohama City University Medical Center, Yokohama, Kanagawa, Japan, 2 Ultrasound Department, The First Affiliated Hospital of Xi'an Jiaotong University, Xi'an, Shaanxi, People's Republic of China, 3 Radiation Oncology Center, Ofuna Chuo Hospital, Kanagawa, Japan, 4 Division of Gastroenterology, Yokohama City University Graduate School of Medicine, Kanagawa, Japan

* kz-numa@urahp.yokohama-cu.ac.jp

**Data Availability Statement:** Detailed data of patient information cannot be shared publicly

## Abstract

### Aim

To evaluate the safety and efficacy of the administration of radiofrequency ablation (RFA) and stereotactic body radiotherapy (SBRT) in the short term to the same patients in Barcelona Clinical Liver Cancer (BCLC) stages 0–B1.

### Methods

From April 2014 to June 2019, we retrospectively reviewed BCLC stage 0–B1 patients with fresh hepatocellular carcinoma (HCC) lesions that were repeatedly treated by RFA (control group, n = 72), and by RFA and subsequent SBRT (case group, n = 26). Propensity score matching (PSM) was performed to reduce the selection bias between two groups. Recurrence, survival, Child–Pugh scores and short-term side effects (fever, bleeding, skin change, abdominal pain and fatigue) were recorded and analyzed.

### Results

After PSM, 21 patients remained in each group. Seventeen and 20 patients in the case and control groups experienced recurrence. For these patients, the median times to progression and follow-up were 10.7 and 35.8 months, respectively. After PSM, the 1-year progression-free survival rate in case and control groups were 66.7% and 52.4%, respectively (P = 0.313). The inter-group overall survival (OS) was comparable (3 and 5-year OS rates in case groups were 87.3% and 74.8%, while rates in control groups were 73.7% and 46.3%, respectively; P = 0.090). The short-term side effects were mild, and the incidence showed

because of the privacy of patients. Data are available from the Ethics Committee of Yokohama City University Medical Center (contact via https://yokohama-cu.bvits.com/rinri/login.aspx?ReturnUrl=%2frinri%2fCommon%2f) for researchers who meet the criteria for access to confidential data and electronic medical record system of Yokohama City University medical center.

**Funding:** The author(s) received no specific funding for this work.

**Competing interests:** The authors have declared that no competing interests exist.

no inter-group difference. The 1-year rates of the Child–Pugh score deterioration of $\geq 2$ in case and control groups were 23.8% and 33.3% ($P > 0.05$), respectively.

## Conclusion

The short-term administration of RFA and SBRT to the same BCLC stage 0–B1 patients may be feasible and effective because of their good prognosis and safety.

## Introduction

Patients in Barcelona Clinical Liver Cancer (BCLC) stages 0–B1 have well-preserved liver function and are expected to receive curative treatment [1]. Radiofrequency ablation (RFA) is regarded as a favorable or even first-line treatment option for inoperable hepatocellular carcinoma (HCC) at these stages [1, 2]. However, there are many conditions for which RFA is unable to be performed or would predictably have an inferior outcome; for example, a risky location of lesions (lesions adjacent to vessels, diaphragm or heart or protruding from the hepatic surface), invisibility on ultrasound and lesions with a large size [3]. Stereotactic body radiotherapy (SBRT) is not recommended as the therapy for early-stage HCC. However, it has been frequently used in early-stage HCC patients with a reported good prognosis. The overall survival (OS), progression-free survival (PFS) and tumor response of SBRT are comparable with RFA [3, 4]. In addition, for tumors with a size of over 2 cm, SBRT outperformed RFA in local PFS. For lesions of all sizes, SBRT showed a relatively higher 1-year OS and lower incidence of acute grade 3+ complications [5]. In view of the above reported facts, on the premise of the priority of RFA as the primary treatment for BLCL stage 0–B1 patients with inoperable HCCs, we aimed to introduce SBRT as a novel alternative treatment strategy for lesions that were ineligible for RFA treatment. RFA and subsequent SBRT treatment included multifocal lesions treated with RFA and SBRT separately, and local/distant recurrent lesions after RFA treatment were treated with SBRT. For the latter case, a few studies exhibited the feasibility of using SBRT on residual RFA-treated lesions with encouraging outcomes [6, 7]. Regrettably, however, there are no data available on the conditions of the short-term application of RFA and SBRT in the same patient for multifocal lesions or intrahepatic distant recurrence, which is very common in the progress of HCC. The survival benefits obtained from the additional treatment of SBRT after initial RFA need to be weighed against the relative possible risks of the procedure-related cumulative toxicities of the two modalities. In other words, both the feasibility and safety of this novel treatment strategy should be evaluated in detail before it is put into clinical practice.

To address this issue, in the present study, we investigated the outcomes and safety of performing RFA and SBRT methods in the short term. We made a comparison with the already well-recognized therapy of repeated RFA [8] in order to determine the feasibility and effectiveness of our novel RFA and SBRT therapy for BLCL stage 0–B1 patients.

## Materials and methods

### Patient enrollment

From April 2014 to June 2019, we retrospectively reviewed patients with BCLC stages 0–B1 with fresh HCC lesions. According to the Japan Society of Hepatology Guideline [9], the diagnosis of HCC was either based on typical radiological criteria or proven by histopathology. All

the RFA procedures were performed at Yokohama City University Medical Center (YCUMC), while SBRT treatment was undertaken at Ofuna Chuo Hospital. Notably, the first diagnoses of HCC of these patients were done in YCUMC. If the patients needed to be treated with SBRT, their receiving doctors at YCUMC would contact Ofuna Chuo Hospital and refer the patients to Ofuna Chuo Hospital for SBRT treatment. Clinical information (such as gender, age, etiology of HCC, Child–Pugh classification), imaging data, histology reports and treatment processes were retrospectively collected from a review of the electronic medical- recording system, the radiology database and pathology records from YCUMC, respectively. Data collection and the analysis of all enrolled patients and lesions in our retrospective study were approved by the institutional review board of YCUMC (the original approved number was "B180200054" and the updated number is "B200300052") and were in compliance with the principles of the Declaration of Helsinki; the requirement for informed consent was waived. The information from all patients was fully anonymized. When downloading, copying and using the patient's information, a CD-R or USB rather than a network was used, and a password was set for the database.

First, we included the patients in a general policy: (1) in BCLC stage 0–B1, patients were supposed to receive curative rather than palliative treatment; (2) the primary treatment was RFA; (3) the next treatment (multifocal HCC lesions in the same patients or local/intrahepatic recurrence after the initial RFA) would be either RFA or SBRT; and (4) the time interval between the primary treatment (RFA) and the next treatment (RFA or SBRT) would not be more than three months.

The case group and the control group were not randomly assigned. If RFA was predicted to be performed well, RFA was preferred as the first choice. Otherwise, SBRT therapy was performed instead of RFA. The case group had to satisfy the conditions that the subsequent treatment in a short period (within three months) would be SBRT, regardless of the recurrence of RFA-treated lesions or another fresh lesion. The control group enrolled patients with lesions, all of which were treated by RFA in a short period, including multifocal lesions, recurrent RFA-treated lesions and other fresh lesions. The selection of the study population is presented in a flow chart below (Fig 1).

## Treatment modalities

**RFA procedure.** As Wang F's study described [10], RFA was performed using a 480 kHz generator (VIVA RF generator; STARmed, Gyeonggi, Korea) and a 17-gauge, internally cooled, adjustable RF electrode (Proteus; STARmed, Gyeonggi, Korea). The lengths of the active tips of the electrodes were selected mainly according to the location and size of the lesion. For example, for lesion with a diameter of no more than 2 cm, an electrode with a 2 cm tip was selected and ablated at 20 W. Otherwise (for sizes larger than 2 cm), an electrode with a 3 cm tip and ablation at 40 W was selected. The median duration and temperature of one ablation was 12 minutes and maintained above 60˚C, respectively. Post-operative contrast-enhanced ultrasound (CEUS) examination was undertaken to evaluate the adequacy of ablation. A complete ablation was defined as no perfusion of contrast agent into the ablative area (which completely covered the lesion area as a whole), showing a completely black appearance with a distinct boundary. When the ablation was completed, the needle tip was kept hot when the needle was retracted to prevent bleeding from thermal coagulation, seeding along the puncture route.

**SBRT procedure.** The process was performed according to Takeda A's study [11]. Multi-arc, dynamic conformal radiation was planned using a radiation treatment planning system (FOCUS XiO, version 4.2.0–4.3.3: Computerized Medical Systems, St Louis, Mo) and was performed using x-rays from a 6 MV linear accelerator (Clinac 2100C; Varian Medical Systems

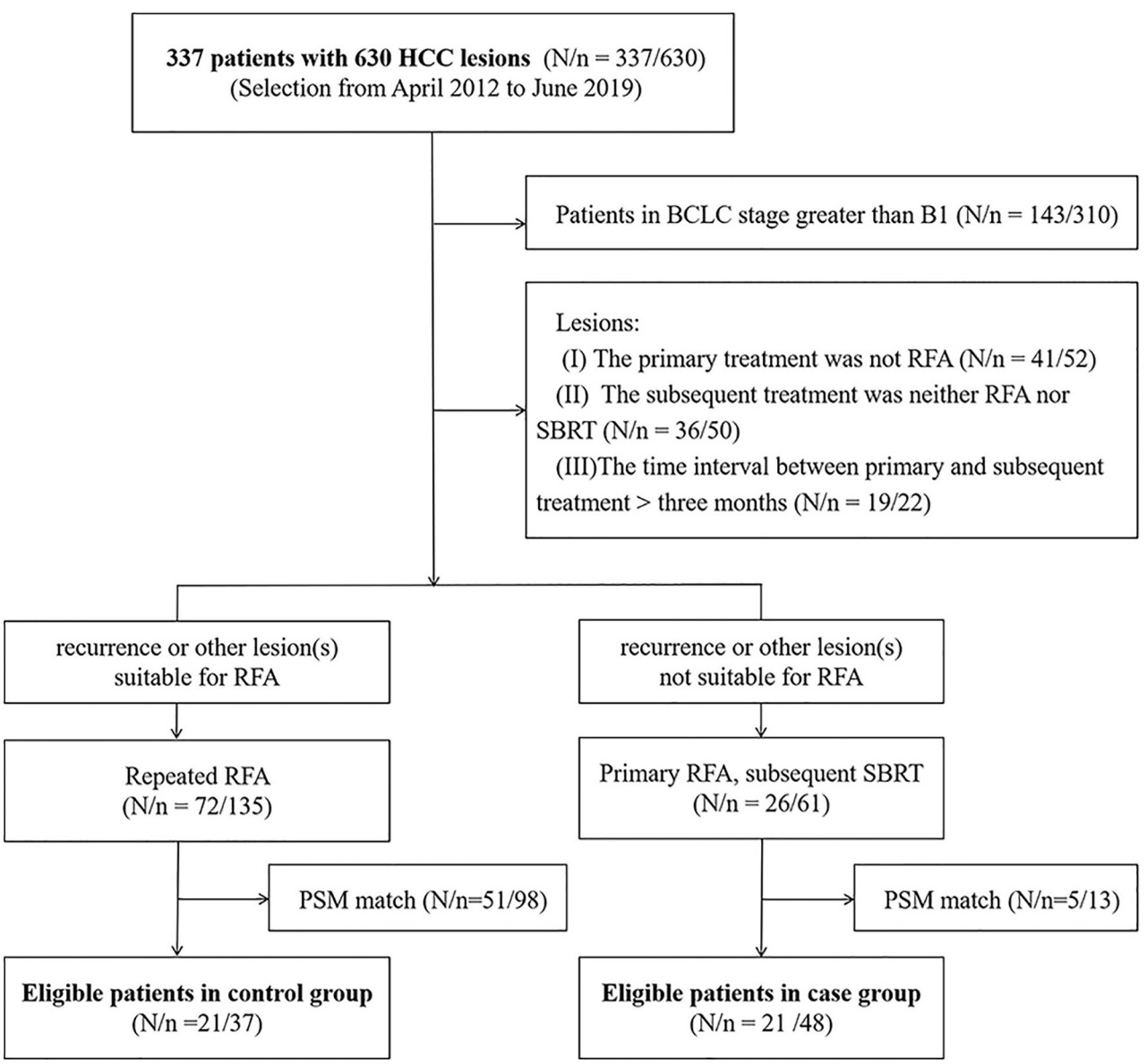

**Fig 1. Flowchart of the study population.** In total, 48 lesions from 21 patients in the case group and 37 lesions from 21 patients in the control group, respectively, were finally used for data analysis after PSM. PSM: propensity score matching; RFA: radiofrequency ablation; SBRT: stereotactic body radiotherapy; HCC: hepatocellular carcinoma; BLCL: Barcelona Clinical Liver Cancer.

Inc, Palo Alto, Calif). Generally, SBRT procedures with total doses of 35 Gy were delivered in five fractions over 5–7 days. Notably, for lesions which were close to the gastrointestinal tract, the dose for a 10 cc area of the gastrointestinal tract was limited to <25 Gy. In this context, the treatment strategy was hypofractionated radiotherapy: a total dose of 36–45 Gy in 12–15 fractions over 16–21 days. Treatment was planned to enclose the planning target volume with a maximal dose of 60–80% of isodose line.

## Follow-up and evaluations of treatment outcomes

The endpoint of overall survival assessment was determined as the patient's death, loss in follow-up or most recent imaging examination of any kind (US, CEUS, CT or MRI). Local

recurrence was defined when a radiological enhancement/ hypervascularity was detected at the treated ablated sites of RFA and a 95% isodose line of SBRT. Intrahepatic recurrence was defined as a newly detected lesion outside of the ablation site (RFA) or planning target volume (SBRT). Follow-up times were calculated from the start date of the prior treatment within this study period, even in cases with previous treatment history for HCC. Short-term side effects of RFA and SBRT were evaluated from the day to one week after treatment. Events of side effects were named and graded according to the National Institutes of Health-defined Common Terminology Criteria for Adverse Events. To assess treatment-related effects on liver function, we analyzed Child–Pugh scores after the latter treatment (RFA for the control group and SBRT for the case group) in one, three and six month and one year follow-ups. The changes compared with baseline Child–Pugh scores were recorded.

## Statistical analyses

Propensity score matching (PSM) was performed to reduce potential confounders and selection biases between the case group and the control group. Propensity scores were calculated using a multivariable logistic regression model including sex, baseline age (with a median value of 70 years-old as a cutoff value), viral etiology (hepatitis B virus (HBV) and/or hepatitis C virus (HCV) versus non-hepatitis B non-hepatitis C (NBNC)), Child–Pugh class (A versus B), pre-treatment alpha-fetoprotein and albumin (normal versus abnormal) and tumor size (with 3 cm as a cutoff value). Following the estimation of propensity scores, the tumors were matched using the 1:1 nearest-neighbor matching algorithm with an optimal caliper of 0.02 and without replacement. Non-matching results were discarded.

After PSM, to compare differences in the baseline characteristics and parameters of outcomes between the two groups, the Pearson's Chi-square or Fisher's exact test was used to analyze categorical variables and the Mann–Whitney U test (non-normally distributed data) or Student's $t$-test (normally distributed data) was employed to compare continuous variables. All statistical analyses were two-sided, and values of $P < 0.05$ were considered statistically significant. Analyses were performed using the statistical software SPSS 24.0 (Inc, Chicago, IL). The survival curves were depicted using the Kaplan–Meier method. Differences of PFS and OS were calculated using log-rank tests.

## Results

### Baseline characteristics

Before PSM, 98 consecutive patients with 196 lesions fulfilled the eligibility criteria and entered the study. Apart from the HCC history, other baseline characters revealed no differences between the two groups. After PSM, 21 patients in each group were retained for afterwards analysis. Eighty-five target fresh HCC lesions from 42 patients were diagnosed between June, 2013 and June, 2019. All the baseline characteristics showed no difference between groups. Baseline characteristics before and after PSM are summarized in Table 1.

### Recurrence and survival after PSM

The endpoint of follow-up was August 20, 2020. The median follow-up time was 35.8 months (range, 2.6–80.5 months) for all patients, and almost equal between the case group (the value of the mean and standard deviation was "40.0 ± 19.7" months) and control group (40.0 ± 22.9 months) ($P = 0.091$). The median time to progression was 10.7 months (with a range of 1.1– 62.2 months) for all patients. The case group (13.9 ± 12.3 months) have slightly longer time to progression than the control group (8.3 ± 9.2 months), however, there was no statistical

**Table 1. Baseline characteristics of patients and tumors before and after PSM[1].**

| | Before PSM | | | | After PSM | | | |
|---|---|---|---|---|---|---|---|---|
| Parameters | Control group | Case group | Total | *P* | Control group | Case group | Total | *P* |
| **No. of patients/lesions** | 72/135 | 26/61 | 98/196 | / | 21/37 | 21/48 | 42/85 | / |
| **BLCL stage (0/A/B1)** | 15/52/5 | 6/17/3 | 21/69/8 | 0.717 | 6/14/1 | 4/14/3 | 16/22/4 | 0.493 |
| **Child–Pugh (A/B)** | 64/8 | 25/1 | 89/2 | 0.272 | 20/1 | 20/1 | 40/2 | 1.000 |
| **HCC history (No/yes)** | 26/46 | 18/8 | 44/54 | 0.004 | 10/11 | 16/5 | 26/16 | 0.057 |
| **Age, mean ± S.D. (year)** | 71.4 ± 9.4 | 68.3 ± 11.7 | 10.6 ± 10.1 | 0.171 | 68.1 ± 10.4 | 69.5 ± 10.6 | 68.8 ± 10.4 | 0.672 |
| **Gender (Male/female)** | 58/14 | 19/7 | 77/21 | 0.426 | 17/4 | 16/5 | 33/9 | 0.707 |
| **Etiology (HCV/HBV/NBNC)** | 55/5/12 | 19/1/6 | 74/6/18 | 0.686 | 13/2/6 | 16/1/4 | 29/3/10 | 0.593 |
| **AFP, mean ± S.D. (ng/mL)** | 125.3 ± 400.4 | 72.1 ± 152.4 | 111.1 ± 352.0 | 0.512 | 18.8 ± 41.4 | 50.7 ± 121.8 | 34.8 ± 91.3 | 0.267 |
| **Albumin, mean ± S.D. (g/dL)** | 3.8 ± 0.6 | 4.0 ± 0.6 | 3.8 ± 0.6 | 0.065 | 3.9 ± 0.4 | 4.0 ± 0.6 | 4.0 ± 0.5 | 0.793 |
| **Lesion number (1/2/(3–5))** | 11/47/14 | 5/14/7 | 16/61/21 | 0.582 | 7/12/2 | 5/9/7 | 12/21/9 | 0.170 |
| **Types of treatment (single lesion/multi-lesions)[2]** | 11/61 | 5/21 | 16/82 | 0.640 | 7/14 | 5/16 | 12/30 | 0.495 |
| **Lesion size, mean ± S.D. (mm)** | 16.3 ± 5.5 | 16.7 ± 9.0 | 16.4 ± 6.6 | 0.778 | 17.1 ± 5.7 | 17.5 ± 9.7 | 17.3 ± 7.9 | 0.847 |

[1] PSM: propensity score matching; BLCL: Barcelona Clinical Liver Cancer; HCV: hepatitis C virus; HBV: hepatitis B virus; NBNC: non-hepatitis B non-hepatitis C; AFP: alpha-fetoprotein.

[2] A single lesion means both that primary and secondary therapy were performed on the same lesion. The latter treatment (SBRT or RFA) aimed to treat the LTP or RFA-incomplete treated lesions. Multi-lesions show that primary and secondary therapy were performed on the different lesions. The latter therapy (SBRT or RFA) aimed to treat the IDR lesion or multifocal lesions that were entirely treated with RFA or separately with RFA and SBRT.

difference between groups (*P* = 0.105). As a whole, 17 patients in the case group (six exhibited local tumor progression while 11 exhibited intrahepatic tumor recurrence) and 20 of 21 patients in the control group (five exhibited local tumor progression while 15 exhibited intrahepatic tumor recurrence) experienced recurrence. The 1- and 2-year PFS rates in the case groups (66.7% and 31.4%) were slightly higher than the control group (52.4% and 28.6%) but demonstrated no statistical significance (*P* = 0.313) (Fig 2c). The 1, 3 and 5 year OS rates in the case groups were 95.2%, 87.3% and 74.8%, and thus comparable with those in the control groups, at 90.5%, 73.7% and 46.3%, respectively (*P* = 0.090) (Fig 2d). The 1 year cumulative intrahepatic recurrence rates in case and control groups were 33.3% and 29.5%, respectively (*P* = 0.968) (Fig 3c). In a 1 year follow-up, there was no local recurrent lesion in the case group, while 25.7% patients in the control group experienced local recurrence (*P* = 0.064) (Fig 3d). The cumulative numbers of HCC related-deaths (e.g., liver failure, multiple organ metastasis and failure) during the entire follow-up were 1 and 4 for the case group and the control group, respectively.

## Toxicities after PSM

Within 1 year of follow-up after treatment, the cumulative rates of a Child–Pugh score deterioration ≥2 in case and control groups were 23.8% (five of 21 patients) and 33.3% (seven of 21 patients) (*P* > 0.05), respectively (Table 2). For the case group, there was no presence of radiation-induced gastrointestinal disorders such as bleeding, nausea and vomiting. One week after SBRT treatment, fatigue was reported in only one patient. The types of short-term side effects caused by RFA treatment were fever, bleeding of the operating area, skin erythema and edema, and abdominal pain. All of these side effects were mild and were relieved after observation or symptomatic treatment. There were no statistical differences between groups (*P* > 0.05) (Table 3). There were no grade 3+ adverse events in both groups.

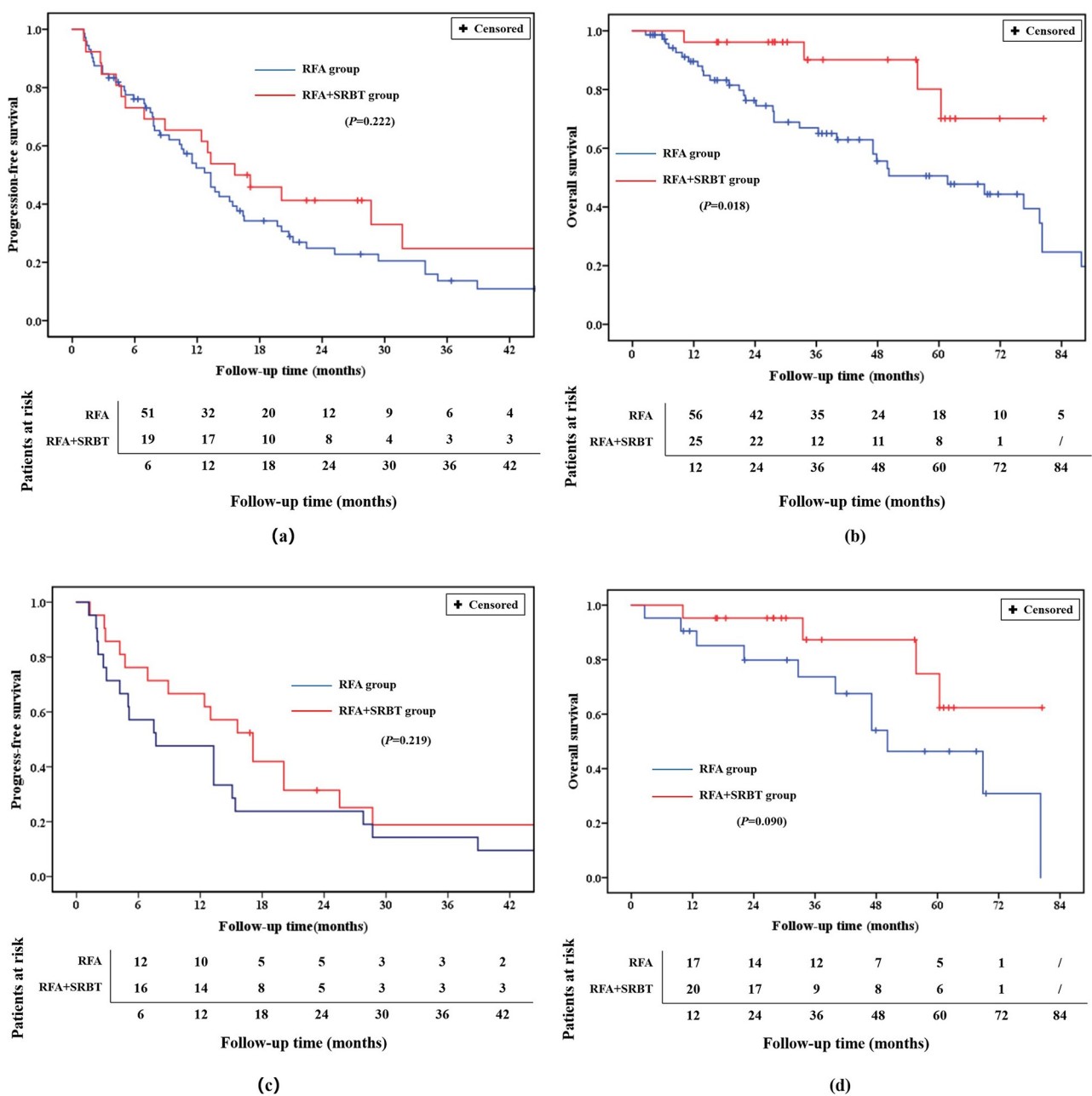

**Fig 2. Progression-free and overall survival rate curves for both groups.** (a) Progression-free survival before PSM. (b) Overall survival before PSM. (c) Progression-free survival after PSM. (d) Overall survival after PSM.

## Discussion

Previous studies have demonstrated that the significant risk factors for recurrence mainly include the size of HCC lesions (especially sizes >3 cm), etiology, serum albumin levels and serum alpha-fetoprotein, while factors affecting survival include the patient's age and Child–Pugh stage [12–15]. Patient age is also considered to be related to post-treatment complications and disease morbidity [15, 16]. In that case, we performed a PSM analysis to reduce potential confounders and selection biases between the case group and the control group.

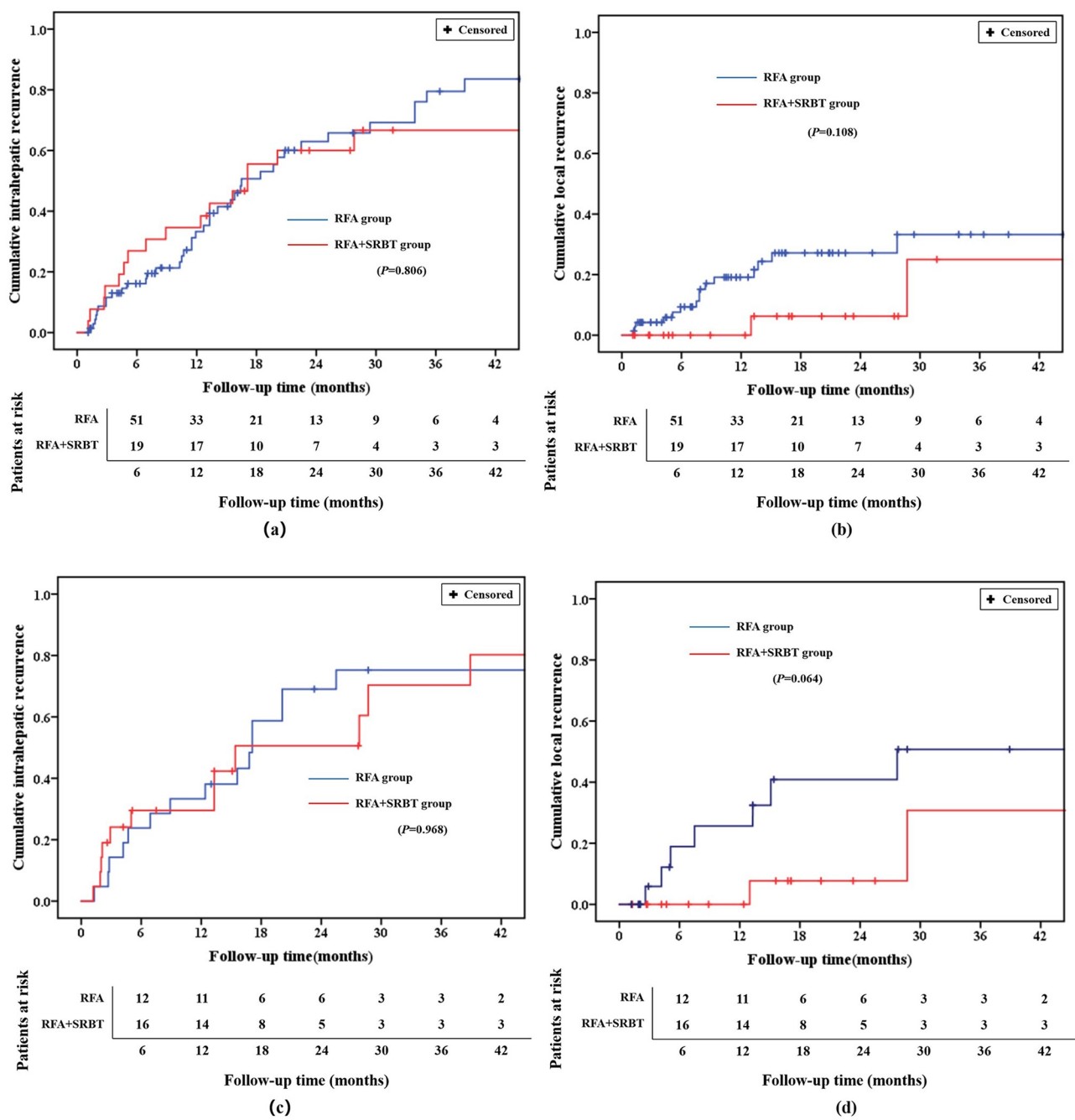

**Fig 3. Incidence curves of cumulative intrahepatic and local recurrence for both groups.** (a) Cumulative incidence of intrahepatic recurrence before PSM. (b) Cumulative incidence of local recurrence before PSM. (c) Cumulative incidence of intrahepatic recurrence after PSM. (d) Cumulative incidence of local recurrence after PSM.

After the PSM and baseline characteristics analysis of the enrolled patients and lesions, we managed to ensure that the inter-group difference of prognosis in afterwards statistics was caused by different treatments rather than the above risk factors.

As a locoregional treatment option, RFA is primarily recommended with curative intent for BLCL stage 0–A patients who have inoperable lesions and are not transplant candidates

**Table 2. Change of Child–Pugh scores in follow-up after PSM[1].**

| Groups | Follow-up time | Unchanged | 1 | 2 | 3 | 4 | -2 | -1 |
|---|---|---|---|---|---|---|---|---|
| **Total patients** | One month | 35 | 5 | 0 | 1 | 0 | 1 | 0 |
| | 3 months | 31 | 8 | 0 | 1 | 0 | 1 | 0 |
| | 6 months | 24 | 11 | 4 | 0 | 0 | 1 | 1 |
| | One year | 23 | 9 | 2 | 1 | 1 | 0 | 2 |
| **Case group** | One month | 19 | 2 | 0 | 0 | 0 | 0 | 0 |
| | 3 months | 16 | 5 | 0 | 0 | 0 | 0 | 0 |
| | 6 months | 12 | 7 | 1 | 0 | 0 | 0 | 1 |
| | One year | 10 | 6 | 3 | 0 | 1 | 0 | 0 |
| **Control group** | One month | 16 | 3 | 0 | 1 | 0 | 1 | 0 |
| | 3 months | 15 | 3 | 0 | 1 | 0 | 1 | 0 |
| | 6 months | 12 | 4 | 3 | 0 | 0 | 1 | 0 |
| | One year | 13 | 3 | 1 | 1 | 0 | 0 | 2 |

[1] PSM: propensity score matching. A positive value indicates that, compared with the baseline Child–Pugh score, the Child–Pugh score increased after follow-up, while a negative value indicates a decline.

[2, 17]. Beneficial effects regarding OS and the safety of RFA treatment have been reported in BLCL stage B1 patients [18]. In a national 10 year survey in Japan of 20,659 HCC cases at BLCL stage A–B by RFA therapy, the 1, 3 and 5 year survival rates were 90.7–97%), (64.7–82.3%), and (43.6–63.8%), respectively [19]. The OS rate yielded from repeated RFA treatment in our study (1, 3 and 5 year OS rates were 90.5%, 73.7% and 46.3%, respectively) was consistent with the national survey data, consistently suggesting that the outcome of RFA treatment is considerable and stable. Nevertheless, in addition to the beneficial effects of OS and safety, the relative high recurrence rate and real demonstration of treatment failure (although with a low incidence rate) for RFA should also be taken seriously and dealt with appropriately. After the first RFA treatment for fresh HCCs, the rate of treatment failure was reported to be 1.2%. For the remaining 98.8% of lesions with complete responses, 6.2% of local recurrence and 35% of intrahepatic recurrence were detected [19]. There are many factors of patients and lesions that might contribute to an insufficient and/ or risky ablation, even for skilled operators. High-risk lesion locations (beneath the diaphragm, subphrenic or perivascular areas) are well-recognized factors that hamper the achievement of complete ablation with sufficient ablative margins [20]. Because of the minimally invasive nature of RFA, a bleeding-prone state of patients (with conditions such as hemodialysis requiring continuous use of anticoagulants) would be a contraindication for RFA. A large lesion size is also considered for a risky factor that affects the result of ablation. Large lesions are believed to more frequently have irregular borders, along with satellite lesions [21], and are thus not

**Table 3. Side effects related to RFA treatment after PSM[1].**

| Side effect | Control group (n = 21) | Case group (n = 21) | Total | P |
|---|---|---|---|---|
| **Fever** | 7 (33.3%) | 6 (28.6%) | 11 | >0.05 |
| **Bleeding** | 5 (23.8%) | 1 (4.8%) | 5 | >0.05 |
| **Skin erythema, edema** | 4 (19.0%) | 0 (0.0%) | 2 | >0.05 |
| **Abdominal pain** | 6 (28.6%) | 6 (28.6%) | 12 | >0.05 |

[1] RFA: radiofrequency ablation; PSM: propensity score matching.

easy to completely ablate even after undertaking multiple RFA sessions. In addition, RFA performance-reduced complete necrosis was reportedly surprisingly high in 53.1% of HCC lesions smaller than 3 cm and 14.3% of lesions larger than 3 cm [22]. A recurrence might potentially be caused by incomplete necrosis and consequently lead to the presence of a residual tumor. For the above situation, which is not eligible for repeated RFA, no matter whether for recurrent lesions or lesions coexisting with RFA-treated lesions, an alternative modality other than RFA should be considered.

SBRT is usually indicated in patients with 1–3 lesions with a maximum diameter of 5–6 cm [23] (which is in agreement with BLCL stage 0–B1) which are not eligible for resection or other local therapies. RFA results in incomplete tumor control with increasing tumor size. In contrast, for tumors <5 cm, there is no size discrepancy in the recurrence for tumors treated with SBRT, and the quality of radiotherapy planning was satisfactory [24]. Unlike RFA, which is greatly restricted by the lesion location, there are very few geographic limitations for thera-peutic approaches according to tumor location in image-guided radiotherapy-adopted SBRT [24] unless the tumor is adjacent to critical structures such as the bowel or heart, etc. A previ-ous study showed significant improved local recurrence in patients treated with SBRT for larger lesions than possible with RFA [8]. Current practice, using SBRT as a sole treatment or in combination with other local therapies (transcatheter arterial chemoembolization, resec-tion), irrespective of small retrospective cohorts or larger series and well-designed phase II tri-als, has consistently confirmed the value of SBRT in the treatment of HCCs by encouraging local control (1 year, 65–100%), OS rates (1 year, 32–94%) and low toxicity [14]. In line with the previous study, our novel therapy of administering short-term RFA and SBRT exhibited an acceptable PFS rate in one year (66.7%) and a stable and high OS rate in continuing follow-up years (the 1, 3 and 5 year OS rates in case groups were 95.2%, 87.3% and 74.8%). In particular, our new therapy showed no local recurrence in the first follow-up year while the repeated RFA treatment yielded a 1 year local recurrence rate of 25.7%. The fact that SBRT is superior to RFA in local recurrence has also been confirmed in much of the literature [4, 5, 25].

Although the prognosis for therapy with RFA and subsequent SBRT is comparable to or even better than that of repeated RFA, it does not come at the cost of reduced safety. SBRT is non-invasive and thus does not cause collateral thermal damage to adjacent structures. The most important dose-limiting factor of SBRT is liver damage. Fortunately, taking the advan-tages of SBRT's characteristics of delivering high individual doses of radiation to target tumors and keeping rapid fall-off doses to surrounding normal tissues [26], there was no significant deterioration of liver function when compared with the control group (the patients' Child–Pugh score deterioration rates of ≥2 in case and control groups were 5 and 7, respectively). In the case group, RFA was performed fewer times than in the control group. It is easy to under-stand the lower incidence of RFA-related complications than the cumulative incidence caused by repeated RFA performance in the control group. The incidence of radiation-related compli-cations was low and almost negligible. Only one patient complained of fatigue, which was con-sidered to be a typical syndrome of radiation hepatitis. Due to strict dose–volume constraints and proper patient selection, most previous SBRT studies have demonstrated a minimal risk of radiation hepatitis (a reported rate was 2.4%) [27]. No severe complications were encountered during the entire follow-up.

This study has several limitations. Firstly, our research has a retrospective nature, with a small number of patients and lesions. Therefore, selection bias was inevitable despite a PSM analysis. In other words, the patients and lesions in this study may not be representative of the HCC population in BLCL stage 0–B1. Secondly, multiple lesion types were treated with SBRT. For example, some lesions showed local recurrence or intrahepatic recurrence after RFA treat-ment, while others to which SBRT was applied co-existed with lesions treated by RFA. Due to

the small number of cases, we were unable to conduct a subgroup analysis, so it is impossible to know whether the outcomes of RFA and SBRT were different in these different settings. Finally, our study was underpowered. Survival curves show that the study group had higher PFS and OS than the control group, yet there was no statistically significant difference. Additional retrospective chart reviews or—better yet—prospective, multicenter, large-scale trials are needed to determine whether the trends we found are in fact statistically significant.

## Conclusion

In patients in BCLC stages 0–B1 with inoperable lesions and who are ineligible for RFA, SBRT may be a promising curative treatment option because of its comparable or even better prognosis and safety than the repeated use of RFA. Our novel therapy regarding initial RFA and subsequent SBRT treatment might provide new ideas for the curative treatment of HCC, but large-scale prospective studies are required before they can be definitively applied in clinical practice.

## Supporting information

**S1 File. The operation details of RFA and SBRT.**
(DOCX)

## Author Contributions

**Conceptualization:** Yuichirou Tsurugai.

**Data curation:** Katsuaki Ogushi.

**Formal analysis:** Feiqian Wang, Hiroyuki Fukuda.

**Investigation:** Makoto Chuma, Shin Maeda.

**Methodology:** Atsuya Takeda, Koji Hara.

**Project administration:** Kazushi Numata.

**Resources:** Makoto Chuma.

**Supervision:** Takahisa Eriguchi.

**Writing – original draft:** Feiqian Wang.

**Writing – review & editing:** Kazushi Numata, Atsuya Takeda.

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
