## [Decision Letter · Decision Letter 0]

27 Nov 2020

PONE-D-20-33979

Safety and efficacy study: short-term application of radiofrequency ablation and stereotactic body radiotherapy for barcelona clinical liver cancer stage 0–B1 hepatocellular carcinoma

PLOS ONE

Dear Dr. Numata,

Thank you for submitting your manuscript to PLOS ONE. After careful consideration, we feel that it has merit but does not fully meet PLOS ONE’s publication criteria as it currently stands. Therefore, we invite you to submit a revised version of the manuscript that addresses the points raised during the review process.

We look forward to receiving your revised manuscript.

Kind regards,

Stephen Chun

Academic Editor

PLOS ONE

Journal Requirements:

2. In the ethics statement in the manuscript and in the online submission form, please provide additional information about the patient records/samples used in your retrospective study, including:

a) whether all data were fully anonymized before you accessed them;

b) the date range (month and year) during which patients' medical records/samples were accessed.

3. Please ensure reporting of 1-year progression-free survival in case group vs control group is reported correctly in the abstract.

4. Thank you for including your ethics statement:

"Our retrospective study design was approved by the institutional review board and in compliance with the principles of the Declaration of Helsinkii; the requirement for informed consent was waived."   

Reviewers' comments:

Reviewer's Responses to Questions

**Comments to the Author**

1. Is the manuscript technically sound, and do the data support the conclusions?

Reviewer #1: Partly

Reviewer #2: Yes

Reviewer #3: Yes

2. Has the statistical analysis been performed appropriately and rigorously? 

Reviewer #1: I Don't Know

Reviewer #2: Yes

Reviewer #3: Yes

3. Have the authors made all data underlying the findings in their manuscript fully available?

Reviewer #1: Yes

Reviewer #2: No

Reviewer #3: Yes

4. Is the manuscript presented in an intelligible fashion and written in standard English?

Reviewer #1: No

Reviewer #2: Yes

Reviewer #3: Yes

5. Review Comments to the Author

Reviewer #1: The manuscript requires extensive editing for grammatical errors as well as English language editing. For example, there are fragments in the Introduction and errors in tense in several sentence. Some of the terms used such as "fresh HCC" is not a term commonly used in the medical literature and would be more appropriately described as "previously untreated HCC."

There are inaccurate statements in the introduction misquoting other reports in the literature. For example, the sentence on line 53-54 in the introduction, which states SBRT outperformed RFA in overall survival (incorrect per that reference) and lower incidence of grade 3+ complications (the difference was not statistically significant), is misleading/false. This statement in the Introduction on lines 56-57, "we aimed to introduce SBRT as a novel alternative treatment strategy for lesions ineligible for RFA treatment" is not supported by the design of this retrospective study.

Because of the small sample size, none of the differences between the two groups are statistically significant; however, they are discussed as having differences between the therapies in the Abstract and Discussion.

Reviewer #2: This is a small retrospective study comparing patients with HCC who received SBRT and RFA in a short interval compared to RFA alone. The authors attempted to correct for possible differences between the groups by doing propensity score matching. The results showed that the case group had comparable outcomes and low toxicity.

1. Please try to edit the grammar of this manuscript, as there are several portions that are not written in clear and correct English. For example, the term "RFA and SBRT short-term performing" throughout the manuscript does not make sense grammatically and should be replaced. Other sentences that need to be re-written include: lines 61-64, 66-68, 248-249, 252 (intrahepatic spelling), 254-256, 264-266, 266-268, 294-298, 300-302, 311-312 ("respective"), 321-322.

2. Lines 172-173: You should mention the fact that there was an almost statistically significant difference in "HCC history" between the case and control groups, and perhaps discuss that in the discussion as a possible limitation.

3. Lines 274-277: there should be mention of the fact that SBRT tumor coverage can be limited (and possibly results in compromise in tumor control) if the tumor is adjacent to critical structures such as bowel or heart, etc.

Reviewer #3: I think this manuscript is excellent "as is," however I feel it could be made both more concise and slightly more clear.

Most importantly, to me "retrospectively enrolled" is a phrase I have not heard or read before - please do not use it. I would intend use "retrospectively reviewed" or "retrospective chart review." Did the authors do a retrospective chart review? Did you use an electronic medical record and/or artificial intelligence to select patients? I would like to see more in the methods section about this. I would also like to read more about PSM methods. I think Figure 1 is excellent and should be published as is.

I was glad to see in Methods section that you had IRB approval. Helsinki is spelled with only 1 i at the end in both Finnish and English, however - please correct this typo. I was concerned that you did not have any IRB approval or waiver when I read n/a in the Ethics Statement section - does that section need to be updated at all?

Please correct grammar. E.g. In Introduction page 3 line 46 "many conditions in which RFA is unable... inferior outcome; for example, large size, invisibility on ultrasound, and risky location, including close proximity to vessels, diaphragm or heart, and protrusion from the hepatic surface." There are a few other instances where "is" (singular) or "are" (plural) are used incorrectly, likely due to editing software error, for example page 5 line 89 should be "is not more than three months." I would use "First" instead of "Firstly" (page 5, line 84).

In Table 2, I would consider using - not / just because to me, - is a lot easier on my eyes. Also, please clarify, what is the difference between / (or preferably - ) and 0?

Table 3 could be shortened by eliminating the "no" rows and the "no / yes" column, instead of "Skin Change" label "Skin redness, swelling" or "Skin erythema, edema." All your numbers add up to 100% in each side effect category so it does not appear that a single patient was *not* asked about these side effects, hence the "no" rows being essentially useless. This is the only concern I have is that was every single "no" patient asked about every single side effect, every single time??? I appreciate this "no" data availability to me as a reviewer, but I think it need not be published.

Figures 2 and 3 are excellent. The y-axis labels especially are slightly blurry to me. Please ensure high enough resolution for publication because even on my regular monitor, it is blurry, and enlarging / zooming in only makes it appear more pixelated.

Discussion

Over all, this section could be made more concise. I am focusing my suggestions for improvement on the last paragraph of Discussion, as below:

Page 19 line 310 should be "inevitable despite PSM analysis." I would eliminate the world "well" in line 311. I believe the first word in line 312 should be "perspective" instead of "respective" (or perhaps you meant "retrospective"?) In lines 315-316, you could perhaps note that your study is underpowered to show statistically significant difference, despite the trends, hence the need for multi center, large-scale studies as you noted. In line 318 I would not the lesion types treated with SBRT were multiple, for example "local recurrence or intrahepatic recurrence...lesions treated by RFA."

I recommend re-ordering the last paragraph in the Discussion session as follows:

Keep lines 308 - end of line 311 the same. Then "Also, the lesion types treated with SBRT were multiple, for example... [as I wrote out in the paragraph immediately above]...lesions treated by RFA. Due to the small number of cases, we are unable to conduct a subgroup analysis, so it is impossible to know whether the outcomes of RFA and SBRT are different in these different settings. Finally, our study was underpowered. Survival curves show that the study group had higher PFS and OS than the control group, yet there is no statistically significant difference. Additional retrospective chart reviews or, better yet, prospective, multicenter, large-scale trials are needed to determine whether the trends we found are in fact statistically significant."

Conclusion

page 20 line 326 "tend" - what do you mean??

I would rewrite line 326 "SBRT is a promising and curative treatment option, because of its comparable or even improved prognosis and safety profile, compared to repeated RFA. Our study shows that initial RFA and subsequent SBRT is a safe and efficacious method for curative treatment of HCC, however larger-scale studies are required.

6. PLOS authors have the option to publish the peer review history of their article (what does this mean?). If published, this will include your full peer review and any attached files.

Reviewer #1: No

Reviewer #2: No

Reviewer #3: **Yes: **Liisa L. Bergmann, MD, MBA

---

## [Author Response · Author response to Decision Letter 0]

15 Dec 2020

Editor’s comments

Point: If applicable, we recommend that you deposit your laboratory protocols in protocols.io to enhance the reproducibility of your results. Protocols.io assigns your protocol its own identifier (DOI) so that it can be cited independently in the future. For instructions see: http://journals.plos.org/plosone/s/submission-guidelines#loc-laboratory-protocols

Reply I am quite sorry that I failed to upload our protocols to that identifier (DOI). Therefore, I wrote the steps of RFA and SBRT here as follows.

The operation details of RFA were as follows.

Instruments and equipment

RFA was performed under either grayscale US guidance (for HCC clearly detected by grayscale US) or contrast-enhanced US (CEUS) guidance (for US undetectable HCC). The LOGIQ E9 ultrasound system and a convex probe with a frequency of 1–6 MHz or a micro-convex probe with a frequency of 2–5 MHz were applied. To ablate all the tumors, RFA was performed using a 480 kHz generator (VIVA RF generator; STARmed, Gyeonggi, Korea), capable of producing a maximum power of 200 W and a specific size of 17-gauge internally cooled, adjustable RF electrode (Proteus; STARmed, Gyeonggi, Korea).

Important parameter setting

For CEUS guidance, in general, a low mechanical index (MI) mode (0.2–0.3) was used as the whole liver, which could be observed in real time, allows for repeated observations of the liver in a dynamic real-time manner because microbubbles stay static and undergo less destruction. Unfortunately, deep-located hepatic lesions might invisible with a low MI because of the attenuation of the US beam. If the lesion margin was indistinct (especially if the lesion was isoechogenic) on the grayscale US image and deep-located, high MI mode (0.8–1.2) and/or fusion imaging with enhanced CT/MRI were applied.

Operation tips

1)The lengths of the active tip of the electrodes were 5 mm, 1.0 cm, 1.5 cm, 2.0 cm or 3.0 cm. Each electrode was selected based on tumor size, tumor location, and operator preference. Briefly, for patients with a tumor diameter of 2 cm or smaller, an electrode with a 2 cm tip was inserted into the tumor and ablated at 20 W. For patients with a tumor diameter larger than 2 cm, an electrode with a 3-cm tip and ablation at 40 W was selected. For nodules with a relatively larger size (3–5 cm), the electrode was inserted at different sites.

2)In cases with hypervascular HCCs that exhibited hypervascular enhancement during the arterial phase and hypo-echoic enhancement during the post-vascular phase, we punctured the lesion during the post-vascular phase. In contrast, in cases with hypervascular HCCs that exhibited hypervascular enhancement during the arterial phase and iso-echoic enhancement during the postvascular phase, we punctured the lesion during the arterial phase of CEUS.

Operation steps

All patients received local anesthesia and analgesia before the procedure. In general, for lesion with big size, the first puncture of RFA targeted the center of the largest section of the lesion shown on the US images. Afterwards, the overlapping ablations were performed until the entire lesion was ablated. The ablation algorithm was based on elevations in tissue impedance. The mean duration of one ablation was 12 minutes, and the temperature of the ablated tissue was maintained above 60 ℃. 

Immediately after ablation, the therapeutic response to RFA was evaluated by CEUS or fusion imaging to determine the adequacy of ablation, and whether additional ablation was needed. If sometimes the steam produced by radiofrequency ablation affects the visual field of ultrasound observation, wait for a moment for a clear field of vision before evaluating the adequacy of ablation. For RFA performed under the guidance of CEUS, post-operative CEUS examination was undertaken to see if the edge of the ablation area showed hypervascularity in AP. A complete ablation could be confirmed by no perfusion of contrast agent into the tumor as a whole, showing a “cavity” appearance with a distinct boundary. For RFA treatment guided by fusion imaging, the overlap function of fusion imaging was applied to observe whether the ablative area completely covered the lesion area shown on pre-operative enhanced MRI or CT images. 

After ablation, the needle was retracted, maintaining its tip hot in order to prevent, by thermal coagulation, seeding or haemorrhage along the electrode track. Every procedure was aimed at obtaining a no less than 5 mm safety margin around the treated lesions. 

The operation details of SBRT were as follows.

During free breathing of patients, Spiral, 4-phase, multidetector CT and/or dynamic contrast-enhanced MRI were conducted, and followed by fusion with a slow-scan CT scan (6–10 seconds per slice). The gross tumor volume (GTV), including enhanced tumor, was delineated with the slow-scan CT images. For the internal target volume, an internal margin (4–6 mm) was created around the clinical target volume (CTV) according to the respiratory movement of the diaphragm observed during fluoroscopy. For the planning target volume (PTV), individualized margins of 2 mm were applied around the internal target volume as a setup margin. Multiarc, dynamic conformal radiation was planned using a radiation treatment planning system (FOCUS XiO, version 4.2.0–4.3.3: Computerized Medical Systems, St Louis, Mo) and was performed using X-rays from a 6-MV linear accelerator (Clinac 2100C; Varian Medical Systems Inc, Palo Alto, Calif). Generally, SBRT with total doses of 35 Gy or 40 Gy were delivered in five fractions over 5–7 days. 

For patients with Child-Pugh grade A–B grade, > 20% of the normal liver receiving > 20 Gy, and a total dose of 35 Gy was administered. For other patients, a total dose of 40 Gy were delivered. Notably, for the lesions which were close to the gastrointestinal tract, there was a policy that the dose to a 10-cc area of the gastrointestinal tract should be limited to <25 Gy. In this consideration, the treatment strategy was hypofractionated radiotherapy: a total dose of 36–45 Gy in 12–15 fractions over 16–21 days. Treatment was planned to enclose the planning target volume with a maximal dose of 60%–80% isodose line. 

Journal Requirements

Point 1. Please ensure that your manuscript meets PLOS ONE's style requirements, including those for file naming. The PLOS ONE style templates can be found at https://journals.plos.org/plosone/s/file?id=wjVg/PLOSOne_formatting_sample_main_body.pdf and 

Reply 1: Thank you very much for your kind reminding. We read the formatting sample carefully and made revision of our manuscript in respective of the font, font size and line spacing et. al to abstract, main text and title page accordingly.

Point 2. In the ethics statement in the manuscript and in the online submission form, please provide additional information about the patient records/samples used in your retrospective study, including:

a) whether all data were fully anonymized before you accessed them;

b) the date range (month and year) during which patients' medical records/samples were accessed.

Reply 2: Thank you for your valuable comments to perfect our manuscript. 

a)We added the important information of ethics statement in the main text as follows(page 5, lines 85–99).

All the RFA performance were done at Yokohama City University Medical Center (YCUMC) while SBRT treatment at Ofuna Chuo Hospital. Notably, the first diagnoses of HCC of these patients were done in YCUMC. If the patients need to be treated with SBRT, their receiving doctors of YCUMC would contact Ofuna Chuo Hospital and refer the patients to Ofuna Chuo Hospital for SBRT treatment. Clinical information (such as gender, age, HCC etiology of HCC, Child–Pugh classification), imaging data, histology reports and treatment process were retrospectively collected from a review of the electronic- medical- recording system, the radiology database, and pathology records from our hospital, respectively. Data collection, analysis of all enrolled patients and lesions in our retrospective study were approved by the institutional review board of YCUMC (Original approved number was “180200054”, update number was “B200300052”) and in compliance with the principles of the Declaration of Helsinki; the requirement for informed consent was waived. All patients were fully anonymized of their information. When downloading, copying and using the patient's information, CD-R or USB rather than network were used, and the password was set for the database. 

b)We evaluated the patients enrolled from April 2014 to June 2019 when the target lesions have just been determined as HCC and have not receive any treatment yet. The follow-up duration of these lesions were 2.6 to 98.9 months before PSM and 2.6 to 80.5 months after PSM, respectively. We carefully re-checked our data, the endpoint of follow-up was August 20, 2020. We added “endpoint of follow-up” in the result section of our main text (page 14, line 202).

Point 3. Please ensure reporting of 1-year progression-free survival in case group vs control group is reported correctly in the abstract.

Reply 3: Thank you very much for your valuable comments. We wrote in abstract that “After PSM, the 1-year progression-free survival rate in case and control groups were 66.7% and 52.4%, respectively (P = 0.313).” In result section of main text, we wrote “The 1- and 2-year PFS rate in case groups (66.7% and 31.4%) were slightly higher than control group (52.4% and 28.6%) but demonstrated no statistical significance (P = 0.313)”. 

With careful checking of statistics and description, we believe our descriptions were correct and consistent among the whole main text.

Point 4. Thank you for including your ethics statement: “Our retrospective study design was approved by the institutional review board and in compliance with the principles of the Declaration of Helsinkii; the requirement for informed consent was waived.” 

Reply 4: The full name of the ethics institutional review board was Yokohama City University Medical Center. We have added this important information in the revised main text (page 5, line 93) as follows.

Data collection and the analysis of all enrolled patients and lesions in our retrospective study were approved by the institutional review board of YCUMC (the original approved number was “B180200054” and the updated number is “B200300052”) and were in compliance with the principles of the Declaration of Helsinki; the requirement for informed consent was waived. 

Review Comments

Response to reviewer 1

1-Point 1: The manuscript requires extensive editing for grammatical errors as well as English language editing. For example, there are fragments in the Introduction and errors in tense in several sentence. Some of the terms used such as "fresh HCC" is not a term commonly used in the medical literature and would be more appropriately described as “previously untreated HCC.”

1-Reply 1: We are quite sorry for grammatical errors. According to your suggestion, language presentation was improved with assistance from a native English speaker with appropriate research background. 

1-Point 2. There are inaccurate statements in the introduction misquoting other reports in the literature. For example, the sentence on line 53-54 in the introduction, which states SBRT outperformed RFA in overall survival (incorrect per that reference) and lower incidence of grade 3+ complications (the difference was not statistically significant), is misleading/false. This statement in the Introduction on lines 56-57, "we aimed to introduce SBRT as a novel alternative treatment strategy for lesions ineligible for RFA treatment" is not supported by the design of this retrospective study.

1-Reply 2: Thank you very much for the reviewer’s insight comment. Our original sentence which the reviewer commented was “For tumors with size of over 2 cm, SBRT outperformed RFA in PFS, OS and yielded lower incidence of acute grade 3+ complications.” We referred it from the original research entitled Outcomes After Stereotactic Body Radiotherapy or Radiofrequency Ablation for Hepatocellular Carcinoma. This article reported that for lesions with size of over 2 cm, the HCC lesion group treated by SBRT have higher local PFS than RFA treated group (P=0.025). For lesions with all size, the lesions treated by SBRT showed higher one-year OS (no statistical difference but P value was not given) and yielded lower incidence of acute grade 3+ complications (P=0.31) than that by RFA. 

Notably, we carefully stated “introduce SBRT as a novel alternative treatment strategy for lesions ineligible for RFA treatment” particularly “on the premise of the priority of RFA as the primary treatment for BLCL stage 0-B1 patients with inoperable HCCs”. We did not deny the usefulness and importance of RFA treatment. This statement was not only concluded from reference 5, but also from reference 3 (this article held the view that radiotherapy appears to be an acceptable alternative treatment option for patients who are not candidates for RFA) and 4 (this article held the view that SBRT therapy appears to be an effective alternative to RFA that should be considered when there is a higher risk of local recurrence or toxicity after RFA), which reported a comparable outcome (OS and PFS) of SBRT and RFA treated lesions. 

We realized our expression of reference 5 was not rigorous and slightly incorrect, so we made revisions as follows (page 3, lines 58–59).

In addition, for tumors with size of over 2 cm, SBRT outperformed RFA in local PFS. For lesions with all size, SBRT showed a relatively higher 1-year OS and lower incidence of acute grade 3+ complications [5]. In view of the above reported facts, on the premise of the priority of RFA as the primary treatment for BLCL stage 0–B1 patients with inoperable HCCs, we aimed to introduce SBRT as a novel alternative treatment strategy for lesions ineligible for RFA treatment.

1-Point 3. Because of the small sample size, none of the differences between the two groups are statistically significant; however, they are discussed as having differences between the therapies in the Abstract and Discussion.

1-Reply 3: Thank you very much for the reviewer’s valuable comments. In describing this manuscript, we consistently take an objective view of the values and limitations of our research. In abstract, we described that “after PSM, the 1-year progression-free survival rate in case group (66.7%) were slightly higher than case group (52.4%) (P = 0.313)”. We just showed the difference between two values (66.7% and 52.4%) and particularly put the P value there. We did not write as “a statistical difference”. In results section, we demonstrated that there were higher values of “time to progression” and PFS rate (also some visible trend shown in survival curve) but no statistical inter-group difference in our results. 

In revised limitation part of discussion section, we explained this indifference might due to small sample size and suggested the prospective multicenter, large-scale studies be organized to further verify our conclusion (page 22, lines 345–347). 

Moreover, to make our expression in abstract more rigorous, we changed the original sentence of “After PSM, the 1-year progression-free survival rate in case group (66.7%) were slightly higher than case group (52.4%) (P = 0.313).” into “After PSM, the 1-year progression-free survival rate in case and control groups were 66.7% and 52.4%, respectively.” (page 2, lines 31–32)

Response to reviewer 2

This is a small retrospective study comparing patients with HCC who received SBRT and RFA in a short interval compared to RFA alone. The authors attempted to correct for possible differences between the groups by doing propensity score matching. The results showed that the case group had comparable outcomes and low toxicity.

2-Point 1: Please try to edit the grammar of this manuscript, as there are several portions that are not written in clear and correct English. For example, the term “RFA and SBRT short-term performing” throughout the manuscript does not make sense grammatically and should be replaced. Other sentences that need to be re-written include: lines 61-64, 66-68, 248-249, 252 (intrahepatic spelling), 254-256, 264-266, 266-268, 294-298, 300-302, 311-312 (“respective”), 321-322.

2-Reply 1: We apologize for the language problems in the original manuscript. We carefully checked the entire manuscript for typographic, grammatical and formatting errors by native speaker using the professional English editing service.

2-Point 2. Lines 172-173: You should mention the fact that there was an almost statistically significant difference in “HCC history” between the case and control groups, and perhaps discuss that in the discussion as a possible limitation.

2-Reply 2: Thank you very much for the reviewer’s insight comments. We have noticed the statistical difference of “HCC history” before PSM and described in lines 189–190 as “Apart from the HCC history, other baseline characters revealed no difference between two groups” and in lines 192–193 as “(after PSM) All the baseline characters have no difference between groups.” Most of our statistics analysis, results and discussion are based on the data after PSM, so we did not think the baseline indicator of “HCC history” is a limitation as it showed no inter-group difference.

2-Point 3. Lines 274-277: there should be mention of the fact that SBRT tumor coverage can be limited (and possibly results in compromise in tumor control) if the tumor is adjacent to critical structures such as bowel or heart, etc.

2-Reply 3: We quite agree with the comment and re-wrote the sentence in the revised manuscript as the following (page 20, lines 298–301): 

Unlike RFA performance, which is much restricted by the lesion location, there is no very few geographic limitations for therapeutic approaches according to tumor location in image-guided radiotherapy-adopted SBRT [24] unless the tumor is adjacent to critical structures such as bowel or heart, etc. 

Response to reviewer 3

I think this manuscript is excellent “as is,” however I feel it could be made both more concise and slightly more clear.

3-Point 1: a) Most importantly, to me “retrospectively enrolled” is a phrase I have not heard or read before - please do not use it. I would intend use “retrospectively reviewed” or “retrospective chart review.” 

b) Did the authors do a retrospective chart review? Did you use an electronic medical record and/or artificial intelligence to select patients? I would like to see more in the methods section about this.

c) I would also like to read more about PSM methods. 

d) I think Figure 1 is excellent and should be published as is.

3-Reply 1: a) We quite agreed with replacing “retrospectively enrolled” with “retrospectively reviewed”. We made revision in the main text accordingly (Page 2, line 21). 

b) Clinical information (such as gender, age, HCC etiology of HCC, Child–Pugh classification), imaging data, histology reports and treatment process were retrospectively collected from a review of the electronic- medical- recording system, the radiology database, and pathology records from our hospital, respectively. For better understanding, we added the information in the method section of the main text. We used the classification and filtering function of excel and SPSS rather than artificial intelligence to select suitable patients. 

c) We used SPSS to do the PSM analysis. We did 1:1 matching between the case group and control group (SPSS can only do the most commonly used 1:1 nearest neighbor matching algorithm). We set a cutoff caliper of 0.02 and without replacement, which was much accurate for matching compared with many published articles. We established multivariable logistic regression model including sex, baseline age (70 years-old as a cutoff value), viral etiology (HBV and/or HCV versus NBNC), Child-Pugh class (A versus B), pre-treatment alpha-fetoprotein and albumin (normal versus abnormal), and tumor size (3cm as a cutoff value). These baseline parameters were carefully referred from RFA, SBRT, PSM related published literature of high levels (For example, Koji Hara, et al, 2019, Hepatology; Nalee Kim, et al. 2020, Journal of Hepatology; Masatoshi Kudo, et al, 2019, Cancers). To make it easier for PSM to succeed (as an operating tip for PSM), all the data input the SPSS software for PSM performance should be enumeration data rather than measurement data, which was a pity that would possibly lose some information of the enrolled data. For the 26 patients in case group (RFA+SBRT), 21 were remained after PSM. Of these 21 patients, 13 were perfect matching (with all the enrolled parameters), while 8 were fuzzy matching. Other 5 patients were excluded after PSM because of failure in match. 

d)We are much grateful for Prof. Liisa L. Bergmann’s positive evaluation of our research. 

3-Point 2. I was glad to see in Methods section that you had IRB approval. Helsinki is spelled with only 1 i at the end in both Finnish and English, however - please correct this typo. I was concerned that you did not have any IRB approval or waiver when I read n/a in the Ethics Statement section - does that section need to be updated at all?

3-Reply 2: Thank you very much for Prof. Liisa L. Bergmann’s kind reminding of the spelling of helsinki. We have made revision accordingly. 

In our research, all the RFA performance were done at Yokohama City University Medical Center (YCUMC) while SBRT treatment at Ofuna Chuo Hospital. Notably, the first diagnoses of HCC of these patients were done in YCUMC. If the patients needed to be treated with SBRT, their receiving doctors at YCUMC would contact Ofuna Chuo Hospital and refer the patients to Ofuna Chuo Hospital for SBRT treatment. Data collection and analysis of all enrolled patients and lesions in our retrospective study were approved by the institutional review board of YCUMC and the approval number was B180200054. (In 2018, a few content of the documents for ethical approval has been updated to make it more rigorous and consistent with the actual clinical situation. The updated number was B200300052. The main content was almost the same). We uploaded the documents and materials related to ethical review and approval in attached supplemental documents. Since all the materials are in Japanese, in order to make it easier to understand, we have translated some key sentences in the form of annotations. 

3-Point 3. Please correct grammar. E.g. In Introduction page 3 line 46 "many conditions in which RFA is unable... inferior outcome; for example, large size, invisibility on ultrasound, and risky location, including close proximity to vessels, diaphragm or heart, and protrusion from the hepatic surface." There are a few other instances where "is" (singular) or "are" (plural) are used incorrectly, likely due to editing software error, for example page 5 line 89 should be "is not more than three months." I would use "First" instead of "Firstly" (page 5, line 84).

3-Reply 3: We are terribly sorry for the language problems in the original manuscript. We made revision to the mentioned places. In addition, we carefully checked the entire manuscript for typographic, grammatical and formatting errors by native speaker using the professional English editing service.

3-Point 4. In Table 2, I would consider using - not / just because to me, - is a lot easier on my eyes. Also, please clarify, what is the difference between / (or preferably - ) and 0?

3-Reply 4: Thank you very much for Prof. Liisa L. Bergmann’s valuable comments for perfect our manuscript. There is no difference between “/” and “0” as “/” in Table 2 means there is no such patient (have elevated or declined value of Child-Pugh score). To make it more clear in expression, we changed “/” into “0” in revised Table 2.

3-Point 5. Table 3 could be shortened by eliminating the "no" rows and the "no / yes" column, instead of "Skin Change" label "Skin redness, swelling" or "Skin erythema, edema." All your numbers add up to 100% in each side effect category so it does not appear that a single patient was *not* asked about these side effects, hence the "no" rows being essentially useless. This is the only concern I have is that was every single "no" patient asked about every single side effect, every single time??? I appreciate this "no" data availability to me as a reviewer, but I think it need not be published.

3-Reply 5: We much appreciate Prof. Liisa L. Bergmann’s insight teaching of Table 3. We made revision accordingly as follows. The nurse and/or the resident physician in charge of the patient would ask and check his/her temperature, symptoms and signs as a routine procedure after RFA.

Table 3. Side effects related to RFA treatment after PSM1 

Side effect Control group (n=21) Case group (n=21) Total P 

Fever 7(33.3%) 6(28.6%) 11 >0.05

Bleeding 5(23.8%) 1(4.8%) 5 >0.05

Skin erythema, edema 4(19.0%) 0(0.0%) 2 >0.05

Abdominal pain 6(28.6%) 6(28.6%) 12 >0.05

1 RFA: radiofrequency ablation; PSM: propensity score matching.

3-Point 6. Figures 2 and 3 are excellent. The y-axis labels especially are slightly blurry to me. Please ensure high enough resolution for publication because even on my regular monitor, it is blurry, and enlarging / zooming in only makes it appear more pixelated.

3-Reply 6: Thank you very much for your praise and kind reminding. We checked the submitted Figure 2 and 3. The pixels meet the magazine's requirements of 300 pixels and appeared clear in separated TIF forms. In this revision, we have uploaded a separate TIF table as a supplement.

3-Point 7. Discussion section: Over all, this section could be made more concise. I am focusing my suggestions for improvement on the last paragraph of Discussion, as below:

Page 19 line 310 should be "inevitable despite PSM analysis." I would eliminate the word "well" in line 311. 

I believe the first word in line 312 should be "perspective" instead of "respective" (or perhaps you meant "retrospective"?) 

In lines 315-316, you could perhaps note that your study is underpowered to show statistically significant difference, despite the trends, hence the need for multicenter, large-scale studies as you noted. 

In line 318 I would not the lesion types treated with SBRT were multiple, for example “local recurrence or intrahepatic recurrence...lesions treated by RFA.”

3-Reply 7: We quite agree Prof. Liisa L. Bergmann’s constructive comments to perfect our manuscript. We made revision point-to-point of the above problems in our latest manuscript.

3-Point 8. I recommend re-ordering the last paragraph in the Discussion session as follows:

Keep lines 308 - end of line 311 the same. Then "Also, the lesion types treated with SBRT were multiple, for example... [as I wrote out in the paragraph immediately above]...lesions treated by RFA. Due to the small number of cases, we are unable to conduct a subgroup analysis, so it is impossible to know whether the outcomes of RFA and SBRT are different in these different settings. Finally, our study was underpowered. Survival curves show that the study group had higher PFS and OS than the control group, yet there is no statistically significant difference. Additional retrospective chart reviews or, better yet, prospective, multicenter, large-scale trials are needed to determine whether the trends we found are in fact statistically significant."

3-Reply 8: We much thank Prof. Liisa L. Bergmann’s teaching of the re-organization of our limitation part. The expression of revised limitation part is much clear, rigorous and easy to understand. We learned much and made revision accordingly.

3-Point 9. Conclusion section: page 20 line 326 "tend" - what do you mean??

I would rewrite line 326 “SBRT is a promising and curative treatment option, because of its comparable or even improved prognosis and safety profile, compared to repeated RFA. Our study shows that initial RFA and subsequent SBRT is a safe and efficacious method for curative treatment of HCC, however larger-scale studies are required”.

4-Reply 9: I am very sorry for my inappropriate expression of “purpose” or “aim”. I deleted “tend” and revised that sentence according to Prof. Liisa L. Bergmann’s excellent teaching as follows (page 23, line 350).

In patients in BCLC stages 0–B1 with inoperable lesions and who are ineligible for RFA, SBRT may be a promising curative treatment option... 

Revision by self-check of authors

Apart from reply to the comments of reviewer, we carefully re-checked our manuscript and data again. We made a few revisions to the latest version of submission as follows to perfect our manuscript. 

Additional point 1. For figure 2, we are sorry that the data tables of "patient at risk" of a and c, b and d were mistakenly placed upside down. The revised figure 2 is as follows. 

Additionally, to be more specific and easier for reading, we changed the insert “figure 2” , “figure 3” in the result section of main text into “figure 2c”, “figure 2d”, “figure 3c”, “figure 3d” as follows.

The 1- and 2-year PFS rate in case groups (66.7% and 31.4%) were slightly higher than control group (52.4% and 28.6%) but demonstrated no statistical significance (P = 0.313) (Fig 2c). The 1-, 3- and 5-year OS rate in case groups were 95.2%, 87.3% and 74.8%, comparable with those in the control groups with 90.5%, 73.7% and 46.3%, respectively (P = 0.090) (Fig 2d). The 1-year cumulative intrahepatic recurrence rate in case and control groups were 33.3% and 29.5%, respectively (P = 0.968) (Fig 3c). In a 1-year’s follow-up, there was no local recurrent lesion in the case group while 25.7% patients in the control group confronted local recurrence (P = 0.064) (Fig 3d). 

Additional point 2. As the term of AFP, ALB and TACE have appeared only once in the main text, in principle, they are more suitable to use full name rather than abbreviations. Therefore, we made revision into alpha-fetoprotein and albumin in method section (page 10, lines 172–173); into “transcatheter arterial chemoembolization” in discussion section (page 21, lines 304–305). In addition, we added the full name of AFP in the footnote of Table 1 (page 12, line 189).

Additional point 3. We added “with a median value of” before “70 years-old as a cutoff” in statistical analyses of the method section. We wanted to demonstrate there that this cutoff value is based on median value of all the enrolled patients.

Additional point 4. As we known, without PSM, the results would be inaccurate without balancing the potential effects of baseline factors. Due to the limitation of manuscript space and in order not to confuse the reader with the “before” and “after” PSM results, the data we displayed in the results section are all after PSM performance. To make it clear, we added “after PSM” after the sub-title of “Recurrence and survival” (page 14, line 201) and “Toxicities” (page 16, line 233).

Additional point 5. We checked all the reference we cited. We corrected “BLCL b stage” to “BLCL B stage” in Ref. 1 and “japan” to “Japan” in both Ref. 9 and Ref.18.

---

## [Editor Report · Decision Letter 1]

22 Dec 2020

Safety and efficacy study: short-term application of radiofrequency ablation and stereotactic body radiotherapy for Barcelona Clinical Liver Cancer stage 0–B1 hepatocellular carcinoma

PONE-D-20-33979R1

Dear Dr. Numata,

We’re pleased to inform you that your manuscript has been judged scientifically suitable for publication and will be formally accepted for publication once it meets all outstanding technical requirements.

Kind regards,

Stephen Chun

Academic Editor

PLOS ONE

---

## [Editor Report · Acceptance letter]

26 Dec 2020

PONE-D-20-33979R1 

Safety and efficacy study: short-term application of radiofrequency ablation and stereotactic body radiotherapy for Barcelona Clinical Liver Cancer stage 0–B1 hepatocellular carcinoma 

Dear Dr. Numata:

I'm pleased to inform you that your manuscript has been deemed suitable for publication in PLOS ONE. Congratulations! Your manuscript is now with our production department. 

Kind regards, 

on behalf of

Dr. Stephen Chun 

Academic Editor

PLOS ONE